# A Printable Device for Measuring Clarity and Colour in Lake and Nearshore Waters

**DOI:** 10.3390/s19040936

**Published:** 2019-02-22

**Authors:** Robert J. W. Brewin, Thomas G. Brewin, Joseph Phillips, Sophie Rose, Anas Abdulaziz, Werenfrid Wimmer, Shubha Sathyendranath, Trevor Platt

**Affiliations:** 1Plymouth Marine Laboratory, Plymouth, Devon PL1 3DH, UK; ssat@pml.ac.uk (S.S.); tplatt@dal.ca (T.P.); 2National Centre for Earth Observation, Plymouth Marine Laboratory, Plymouth, Devon PL1 3DH, UK; 3Chatham and Clarendon Grammar School, Ramsgate, Kent CT11 9BB, UK; TBrewin@ccgrammarschool.co.uk (T.G.B.); i7436610@bournemouth.ac.uk (J.P.); i7447085@bournemouth.ac.uk (S.R.); 4Faculty of Science and Technology, Bournemouth University, Bournemouth, Dorset BH12 5BB, UK; 5CSIR-National Institute of Oceanography, Regional Centre Kochi, Kerala 682018, India; anasabdulaziz@gmail.com; 6Ocean and Earth Science, National Oceanography Centre Southampton, University of Southampton, Southampton, Hampshire SO14 3ZH, UK; w.wimmer@soton.ac.uk

**Keywords:** citizen science, 3D printing, water clarity, water colour, secchi disk

## Abstract

Two expanding areas of science and technology are citizen science and three-dimensional (3D) printing. Citizen science has a proven capability to generate reliable data and contribute to unexpected scientific discovery. It can put science into the hands of the citizens, increasing understanding, promoting environmental stewardship, and leading to the production of large databases for use in environmental monitoring. 3D printing has the potential to create cheap, bespoke scientific instruments that have formerly required dedicated facilities to assemble. It can put instrument manufacturing into the hands of any citizen who has access to a 3D printer. In this paper, we present a simple hand-held device designed to measure the Secchi depth and water colour (Forel Ule scale) of lake, estuarine and nearshore regions. The device is manufactured with marine resistant materials (mostly biodegradable) using a 3D printer and basic workshop tools. It is inexpensive to manufacture, lightweight, easy to use, and accessible to a wide range of users. It builds on a long tradition in optical limnology and oceanography, but is modified for ease of operation in smaller water bodies, and from small watercraft and platforms. We provide detailed instructions on how to build the device and highlight examples of its use for scientific education, citizen science, satellite validation of ocean colour data, and low-cost monitoring of water clarity, colour and temperature.

## 1. Introduction

Two of the oldest instruments in optical oceanography are the Secchi disk [1] and Forel Ule (FU) colour scale [2,3]. The Secchi disk is a white disk that is lowered into the water and the depth at which it disappears and reappears from sight is logged. This depth is proportional to the clarity or transparency of the water. The FU colour scale is used to classify the colour of natural waters. It consists of 21 colours ranging from blue to green to yellow to brown, and is used alongside the Secchi disk with the observer typically recording the colour of a submerged Secchi disk (at roughly half the Secchi depth). A detailed history of both the Secchi disk and FU colour scale can be found in works of Wernand [4], Wernand and van der Woerd [5] and Wernand and Gieskes [6].

Despite the establishment of precision optical instruments designed to measure the colour and transparency of water digitally, the Secchi disk and FU colour scale are still frequently used in modern times, owing to their simplicity, low cost and continuity with historical measurements starting back in the 19th century. Measurements of Secchi depth and FU colour have been used to monitor decadal and centennial changes in turbidity, ocean colour and phytoplankton chlorophyll concentration (e.g., [7,8,9,10,11,12,13]). Secchi disk and FU colour theory have also been applied to satellite observations of ocean and lake colour to map the Secchi depth and FU colour at synoptic scales (e.g., [14,15,16,17,18,19,20,21,22]).

Unlike precision electronic instruments, the observer is implicitly connected with a Secchi and FU measurement, since both these measurements are dependent on the psychophysiology of the human eye-brain system and the individual’s perception of colour and contrast [23]. This can cause additional uncertainties, for instance from possible deviations among individuals, yet the direct engagement ensures that the observer has some authentic understanding of the measurement process. The Secchi depth and FU colour scale are thus useful tools for teaching the basic concepts of optical oceanography, teaching the measurement process, and for citizen-science-based projects (e.g., [24,25,26,27,28]). Lottig et al. [24] revealed geographic patterns and long-term trends in lake clarity in the US, based on multi-decadal observations of the Secchi depth, collected voluntarily by citizens. Other citizen science projects have demonstrated the use of citizen Secchi depth and FU observations for applications such as the validation of satellite ocean colour data [26,27,28]. Citizen science projects involving the use of a Secchi disk require the participating groups or individuals either to purchase a Secchi disk or construct their own. Owing to their size (typically 20–30 cm in diameter), conventional Secchi disks can be relatively cumbersome to use.

3D printing is a process in which successive layers of material are laid down onto a surface to form virtually any shape from a digital model. Basic desktop 3D printers can be purchased at low cost (as little as $500 [29]), making desktop computer-aided manufacture possible in any home or academic institution. 3D printers have greatly benefited many areas of science. They have been used in molecular science to investigate and reproduce molecules and structures [30,31], in paleontology and archaeology for reproducing complex skeletons [32,33], in biology for producing human cells and tissue structures [34,35], in chemistry to initiate chemical reactions for chemical synthesis and analysis [36], and in education to engage students to undertake careers in science and technology [37].

Two exciting features of basic 3D printers are: (i) Their ability to manufacture customized scientific instruments that previously needed dedicated facilities to produce [36,38], and (ii) the ability to standardise home-made scientific instruments, previously manufactured using different materials and tools. Three-dimensional printing thus has the potential to revolutionise the design and use of scientific equipment. Whereas commercialised 3D printers vary in cost, technology and capabilities, they are compatible in terms of input formats [39], meaning that precise information on the construction of scientific equipment can easily be disseminated [38]. The use of 3D printers to manufacture customized scientific instruments is apparent in some areas of environmental science (e.g., see [38]), however, it is relatively untapped in limnology and oceanography, despite unprecedented opportunities [40]. For a recent review of applications of 3D printing technologies in oceanography, the reader is referred to the work of Mohammed [40].

In this paper, we present a simple hand-held, pocket-sized device (hereafter denoted the mini-Secchi disk), designed to measure water clarity (Secchi depth) and colour (FU scale) in lake, estuarine and nearshore regions. This standardised device is partly manufactured using a basic 3D printer and made with marine resistant materials. It is inexpensive to produce and easy to use. The size and weight of the device means the instrument is accessible to a range of recreational water users previously not capable of measuring water transparency and colour using conventional instruments (e.g., divers, kayakers, surfers and stand-up paddle boarders). We provide detailed instructions on how to build the device and demonstrate some of its applications.

## 2. Materials and Methods

### 2.1. Mini-Secchi Disk Development

Our objective was to develop a small, light and convenient-to-use Secchi disk and FU colour scale, for use in lake, estuarine and nearshore environments. Our progression from a standard Secchi disk through to the development of the current mini-Secchi disk is illustrated in Figure 1. We started by making a series of prototypes based on the use of retractable tape measures (Figure 1b,c). We then made the decision to reduce the diameter of the Secchi disk from its standard size (30 cm for the ocean, 20 cm for lakes) to 10 cm. This decision was based on two reasons: (i) By making the size of the disk smaller we could make the entire device pocket-sized, and (ii) consideration of the angular subtense of the disk’s radius ($\Phi =r/S_{D}$, where *r* is the disk’s radius and $S_{D}$ the Secchi depth [23]) in lake and nearshore turbid waters. To expand on the latter point, a 30 cm diameter disk is typically employed in the ocean. Here, the average Secchi depth is around 25 m (estimated by application of Equation (17) of Morel et al. [18] to a NASA MODIS-AQUA global chlorophyll-a climatology and taking the global average value for all the Secchi depths so-obtained), which yields a Φ of around 0.006. This value of Φ, in combination with a disk diameter of 10 cm, would lead to a Secchi depth of 8.3 m. Considering the majority of the world’s lakes (and likely much of the world’s estuaries and nearshore waters) have a Secchi depth <8 m [24], it seems reasonable, for the sake of convenience, to use a 10 cm disk in these environments. In other words, for use in lakes and estuaries, where the water is typically more turbid than in the open ocean, it is acceptable to use a Secchi disk of smaller radius than is deployed conventionally in the ocean. Of course, if we were to compare Secchi depth data collected from the mini-Secchi disk with other records collected using different sized disks, we might need to consider a small correction in measurement for a change in disk size, but this would be possible given knowledge of Secchi depth theory [20,23,41]. However, such a correction may not be required always (see discussion of this in Hou et al. [41], who show that the disk size does not significantly alter the Secchi depth).

Having reduced the size of the disk, more prototypes were constructed by attaching a 10 cm disk to a series of small manually-operated tape measures and chalk lines (Figure 1d–f) which were found to operate satisfactorily. With the goal of making the device compact and convenient to use, we tested a series of 3D printed designs (Figure 1g–i), before finally settling on a specific design (Figure 1j).

### 2.2. The Mini-Secchi Disk

A rendering and exploded drawing of the mini-Secchi disk is provided in Figure 2. The mini-Secchi disk is essentially a manually-operated tape measure with a 10 cm white Secchi disk and 100 g brass weight attached to the end of the tape. The weight is manufactured using a metal lathe (or can be computer numerical-control machined) and the Secchi disk is cut out from white polypropylene sheet using a bandsaw or coping saw. The measuring tape, purchasable online, is approximately 8 m long and wraps around a 3D printed bobbin. The tape and bobbin lock into a 3D printed casing. The bobbin is attached to a handle (also 3D printed) that is used to wind the tape in and out of the casing. A polypropylene finger strap and nylon lanyard are attached to the casing, to operate and transport the device safely. A vinyl-laminated Forel Ule colour scale sticker is placed on the outside of the casing (handle-side) and the entire device is fixed together using stainless steel or brass fixings.

The disk, weight and handle of the tape measure are designed to slot neatly into the chassis of the device, making it very compact and convenient to store and transport. The weight has a small O-ring in the centre that ensures it locks neatly into the bobbin. All 3D printed components are made from polylactic acid, a thermoplastic devised from renewable resources or natural starch. As well as having good structural properties, this material is biodegradable and thus can be composted. The stereolithography files (STL) for all 3D printed parts (and the weight) are provided in Appendix A (see Table 1) and can be used for manufacture in most 3D printers. For the mini-Secchi disk, we used Ultimaker 2 and 2+ printers, which were found to be very reliable and robust and relatively low-cost. We used an open source 3D printing software called Ultimaker Cura. The settings were tailored to print more than one part at a time. Cura is continually updating its software. In the latest version the settings can be adjusted to ensure a high-quality print depending on the colour, type of plastic, and quality of plastic you are printing with. There are other 3D printing software available for use, some unique to the brand of printer you use, others open source. When printing, we recommend using the highest print quality you can apply, and if you are making a single mini-Secchi device, to print parts individually. It is important to know your machine and software well prior to printing, and we recommend first running some test prints.

Table 1 provides extensive details on all the components of the mini-Secchi disk: Dimensions, materials used, how each component is manufactured or purchased, and links to all Appendix A required to manufacture the device. Figure 3 illustrates some of the key steps in the manufacturing process.

### 2.3. Operating the Device

To operate the device, the lanyard is slipped over the wrist (Figure 4a) and the weight is pushed out from the bobbin (Figure 4b). Once the disk is detached (Figure 4c), one or two fingers are slipped under the finger strap for support (Figure 4d). The handle is detached from the casing and flipped 180 degrees (Figure 4e), before it is used to wind the tape measure in and out of the casing (Figure 4f). When the measurement is completed, this procedure is reversed and the device can be easily transported and safely stored.

The Secchi depth is measured using standard protocols: The white Secchi disk is lowered into the water and the depths at which it disappears and reappears are recorded (Figure 5a). The Secchi depth is computed by averaging these two depth measurements. It may be feasible to view the depth of disk disappearance and reappearance directly from the measuring tape at the water surface. If this is not feasible, the distance from the hand-held device (casing) to the water surface (DO) can be measured, as well as the total distance (TO) from the device to the depth at which the disk disappears and reappears. The Secchi depth can then be computed by subtracting DO from TO (Figure 5a). Note if the latter is done, it is essential to measure DO accurately and to keep DO constant throughout the measurement (e.g., by maintaining arm at 90 degrees). For accurate Secchi depth readings, the observer should avoid sun glint regions and shadows, ideally conduct the measurement closer to mid-day (or at the very least record the time and location which can be used to compute sun angle), allow their eyes to adapt near to the Secchi depth, write down sky conditions, and repeat measurements to improve precision. The disk must sink vertically though the water for accurate Secchi depth readings. The weight of the mini-Secchi (100 g) should be sufficient for vertical deployment from a fixed platform in waters with low current speed. However, in stronger currents, or in cases where the platform may be moving (e.g., from a boat) extra weight will be required to avoid the disk sinking at an angle. One of the weight designs provided in Appendix A has an attachment where extra weight can easily added in such circumstances (see Figure 6a,b).

The colour of the water is measured by looking at the colour of the Secchi disk at roughly half the Secchi depth, matching it to the closest colour on the colour scale (Figure 5b) and noting the corresponding number. Once the disk is at half the Secchi depth, it is relatively straight-forward to turn the hand holding the device so that your palm is facing up and the scale is visible and you can see the disk (see photo in Figure 5b). After operating the device, the mini-Secchi should be cleaned with fresh water and stored in a dry location. To remove any dirt from the white disk a little washing up liquid may be used. For maintenance, we also recommend occasionally using a little silicon lubricant on the weight, the inside of the bobbin where the weight locks in, and at the end of the measuring tape.

In addition to measuring the Secchi depth and water colour, the mini-Secchi disk may also be used as a platform for deploying other sensors. Figure 6 show two examples of attaching an iButton temperature logger (DS1922L Thermochron D/logger, accuracy of ±0.5 °C, operating range −40 °C to +85 °C) to the weight of the mini-Secchi disk for measuring water temperature. In Figure 6c,d, the iButton is housed in a Thermochron waterproof capsule (DS9107) and attached to the weight of the mini-Secchi disk. In Figure 6e,f, the iButton is housed in cheaper Thermochron sinking silicon capsule (Th-Silenc) which slips onto the standard weight of the mini-Secchi disk. For the latter, and when using the weight with attachment, the Thermochron silicone enclosure with handle (Th-foenc) could also be used, either by sliding the Th-foenc silicon casing over the weight (as in Figure 6f) or by attaching the Th-foenc handle to the weight attachment, similar to Figure 6d. If using these temperature sensors, it is important for the user to quantify the response time of the housed iButton, to allow enough deployment time for the iButton to respond to the water temperature, and to ensure that the housed iButton is shaded by the disk, to minimise direct heating of the housed iButton by sunlight.

### 2.4. Testing Applications of the Mini-Secchi Disk

In this section we describe four exercises that highlight applications of the mini-Secchi disk.

(1) To demonstrate the use of the mini-Secchi disk as a tool for characterising spatial variations in lake clarity and colour, it was deployed alongside a Conductivity, Temperature and Depth (CTD) rosette sampler on a two-day field trip through Vembanad lake in Kerala, India, on the 30th and 31st May 2018. In addition to measuring conductivity (from which salinity was derived) and temperature, the CTD rosette sampler was also equipped with a WET Labs ECO triplet that included a backscattering sensor for measuring water turbidity. Over the course of the two days, and between 08:20 and 16:35 local time (GMT + 5.5 h), surface waters (top 1 m) of 13 stations were sampled spanning the length of the lake and other parts of its wetland system (Figure 7a). The mini-Secchi disk was deployed at each station alongside the CTD, and the Secchi depth and FU colour were recorded.

(2) The size of the mini-Secchi disk means the instrument is accessible to a range of recreational water users previously not capable of measuring the Secchi depth and water colour using conventional instruments. This is demonstrated here in the nearshore using a recreational surfer. Between 28th February and 4th March 2013, a mini-Secchi disk (prototype version with 10 cm diameter disk, Figure 1) was deployed at Playa Langosta in Costa Rica (outside the surf zone) by a recreational surfer during three surfing sessions (Figure 8a,b). The surfer carried the mini-Secchi disk in a small waist bag (bum-bag or fanny-pack). Between 11:00 and 14:00 local time (GMT-6 h), the surfer was instructed to paddle seaward of the surf zone and record the Secchi depth and water colour using the mini-Secchi disk. Each deployment was repeated five to seven times, with the mean and standard deviation of the Secchi depth computed. The FU colour was also logged for each session. As complementary data, wind speed measurements were taken from a nearby weather station (Sede de Guanacaste, Liberia, data extracted from https://www.wunderground.com/). Three-hourly wind speed measurements centred on the time of each surfing session (for the 28th February, and 3rd and 4th March), or centred at local noon for the days of no deployment (1st and 2nd March), were extracted and the average and standard deviation of these measurements were computed. Wave height data were also extracted for Langosta, Costa Rica, from a surf forecast website (Langosta historic forecast, https://www.magicseaweed.com/ pro version) which uses a wave model forced with observations. The average, minimum and maximum wave heights were taken centred on the time of the surfing session, or at local noon for the days of no deployment. Finally, two photographs taken of the surfer surfing on the 28th February and 3rd March were also utilised for qualitative analysis.

(3) To demonstrate the use of the mini-Secchi disk as an educational tool for teaching the concepts of aquatic optics, and to quantify the average deviation in Secchi disk readings between individuals, we constructed a simple experiment that could be conducted in any classroom. A standard-sized indoor dustbin was filled with water and a 10 cm white Secchi disk was placed at the bottom of the bin. We then added drops of dissolvable navy blue food colouring, purchased from a supermarket, while stirring the water until the white disk was no longer visible. We then removed the white disk. Participants were then asked to record the Secchi depth of the dustbin water using the mini-Secchi disk. This experiment was conducted at a scientific meeting of the Challenger Society and Remote Sensing and Photogrammetry Society Marine Optics Special Interest Group on the 16th December 2013 (“A kaleidoscope of colour: From the turbid to the oligotrophic”), where 18 participants took part, and during a design and technology lesson at a secondary school (at Chatham and Clarendon Grammar School) on the 4th July 2014, where 12 participants took part (Figure 9). Although conditions remained constant during each experiment, the two experiments differed in the amount of food colouring used and in the light environment.

(4) To demonstrate that the mini-Secchi disk can be used as a platform for deploying other sensors, we tested the accuracy of temperature measurements collected by attaching an iButton temperature logger to the weight of the mini-Secchi disk (see Figure 6c,d). These tests were conducted on the 28th Atlantic Meridional Transect (AMT) cruise. AMT is a programme that conducts oceanographic research during an annual voyage between the UK and destinations in the South Atlantic. AMT28 departed from the UK on 23rd September 2018 on the *RRS James Clark Ross* and arrived in the Falklands on the 29th October 2018. Twice a day (around pre-dawn and local noon), the ship stopped to conduct vertical oceanographic profiles of various biological, chemical and physical variables using a CTD rosette sampler. During these profiles, a Small Oceanographic floating Device (SOD) was deployed near the CTD rosette. The SOD is a small float with fins integrated for stability and two plugs for attaching rope to either side of the float (Figure 9a). On the topside, a 10 m rope was attached for lowering the SOD into the water. On the underside, there was a 1 m rope with a mini-Secchi disk attached. On the weight of the mini-Secchi disk an iButton temperature logger (DS1922L Thermochron D/logger), housed in a Thermochron waterproof capsule (DS9107), was attached (see Figure 6c,d and Figure 9a). The SOD was deployed for a 20 min period at 31 stations between 50° N and 4° S, spanning a temperature range of nearly 15 °C (Figure 10a). 1-Wire iButton software was used to launch the iButton prior to deployment and to upload temperature data. The first 7.5 min and last minute of data were removed, to discard any data collected when the iButton was coming in and out of the water and to allow plenty of time for the housed sensor to respond to the water temperature. The median of the remaining data was taken as the temperature reading from the iButton. For comparison with the iButton data, during each SOD deployment, we extracted the median water temperature from the ships underway CTD which samples continuously from the ship’s flow-through system at a nominal depth of 5 m, and also from the top 5 m from the profiling CTD (upward profile). According to the manufacturer’s specifications, the DS1922L Thermochron D/logger has an accuracy of ±0.5 °C. To test this and to remove any systematic bias in the loggers temperature measurements, it was calibrated against a NIST-traceable (and NPL-traceable) Hart Scientific 1504 temperature bridge and Themometrics ES 225 temperature probe (accurate to 0.003 °C at 24.85 °C) at two contrasting temperatures in a recirculating water bath [42], three times during AMT28.

## 3. Results and Discussion

### 3.1. Mini-Secchi Disk Applications

#### 3.1.1. Capturing Spatial Variations in Lake Clarity and Colour

Results from the deployment of the mini-Secchi disk and CTD rosette sampler in Vembanad lake are shown in Figure 7. Surface salinity was highest towards the mouth of Vembanad lake (close to its entrance into the Indian Ocean) and decreased steadily towards the south of the lake (Figure 7f). Surface water temperature was lower (∼28 °C) at the mouth of the lake (Stations 1-2) and in the Muvattupuzha river (Station 7); elsewhere it was relatively constant (∼29 °C) (Figure 7e). Clear spatial variations in Secchi depth were found during the field trip (Figure 7b). Water clarity was highest near the mouth of the lake (Stations 1–2) and in its southern half (Stations 8–13). Water clarity was lowest (∼0.5 m) in the central to northern part of the lake (Stations 3–7). Secchi depth was inversely related to water turbidity (Figure 7b,c, r=−0.89, p<0.001) highlighting the use of the Secchi disk as a simple tool for quantifying turbidity. Variations in Secchi depth recorded during the field trip (average 0.84 m, range 0.4–1.3 m) were consistent with previous Secchi depth data collected in the Eramalloor region of Vembanad lake over an 11 month period in 2015–2016 (average 0.71 m, range 0.4–1.1 m; [43]). Forel Ule colour measurements (Figure 7d) indicated the northern part of the lake (Stations 1, 2 and 4), and the Muvattupuzha river (Station 7), were browner in colour than the southern part of the lake (Stations 8–10) during the field trip.

The Secchi disk and Forel Ule colour scale have a proven history in quantifying temporal and spatial variations in lake clarity and colour [22,24]. The data collected in Vembanad lake demonstrate that the mini-Secchi device can be used for collecting these measurements, but with the advantage of being a more convenient tool owing to its small size and compact design. Secchi depth and Forel Ule observations are increasingly being used for verifying remotely-sensed temporal and spatial variations in water clarity and colour [22,26]. They are thought to be particularly useful for ground truthing satellite data in lake regions with limited resources for monitoring, with local communities determined to help preserve the health of the lakes on which their lives are dependent. It is often these regions that are most vulnerable to anthropogenic problems such as eutrophication and to issues like climate change, and where an inexpensive, printable device such as the mini-Secchi disk could be of real use.

#### 3.1.2. Recreational Water User Deployment in the Nearshore

Results from the deployment of the mini-Secchi disk at Playa Langosta in Costa Rica by a surfer are shown in Figure 8. During the five-day period, the surfer observed an increase in the FU colour (from 16 to 18) and a significant reduction in the Secchi depth (from 1.77 m to 0.92 m). The surfer noted that these changes coincided with an increase in wind speed, little change in wind direction (ENE direction, directly offshore) and no change in average wave height. These observations were verified with wind data from a nearby weather station (Sede de Guanacaste, Liberia) and data from a wave model forced with observations (Figure 8b). Furthermore, photographs taken of the surfer on the 28th February (Julian day 59) and 3rd March (Julian day 62) give further qualitative support to these results (Figure 8c), illustrating similar wave heights on these two dates, higher wind speeds on day 62 (see white caps in background of photo not seen in photo on day 59), and suggest an increase in turbidity and a change in colour between these dates (darker and browner waters on day 62). We speculate that the decrease in Secchi depth (increase in turbidity) and change in colour (toward browner, likely sediment dominated water) was caused by wind driven convection resulting in an increased entrainment of sediment from the seabed into the water column.

The surfer example (Figure 8) demonstrates that, owing to its small size and light weight, the mini-Secchi disk is accessible to a range of water users who were previously not capable of measuring the Secchi depth and FU colour using standard instruments. Recreational watersports users are a community who may benifit from using the mini-Secchi disk. Water clarity is important for activities such as diving, and water quality is a concern for all recreational watersports users. It has been estimated that there are as many as 10 million surfers [44] and 10 million SCUBA divers and snorkellers [45] globally. Recently, Brewin et al. [46] made an analysis of a selection of recreational activities that directly interact with the aquatic environment, and estimated 125 million people engage in these activities in the UK and US alone. There have been studies demonstrating that surfers [47,48,49], divers [50,51,52], stand-up paddle-boarders [53], and sailors [54] can contribute significantly to environmental data collection in aquatic waters. Furthermore, locations where these sports take place are important regions for marine biodiversity [55,56,57] and support various ecosystem services [58]. Tapping into recreational watersports using the mini-Secchi disk could help increase spatial and temporal measurements of water clarity and colour in these regions.

#### 3.1.3. A Tool for Teaching Aquatic Optics

Results from the classroom experiments using a dustbin filled with water and food colouring are shown in Figure 9. The first experiment at the Marine Optics Special Interest Group meeting (Figure 9a,b), in which participants were scientists, yielded an average Secchi depth reading of 36.8 cm, a median of 36.5 cm and a standard deviation of 2.7 cm. The second experiment at the secondary school (Figure 9c,d), in which participants were predominately school children, yielded an average Secchi depth reading of 25.8 cm, a median of 26.0 cm and a standard deviation of 1.9 cm. Differences in Secchi depth readings between experiments were expected considering contrasting amounts of food colouring used and alternative light environments. Yet, the percentage deviation (standard deviation divided by the mean multiplied by 100) was very similar between the two experiments (7.3% and 7.4%) and suggests differences in Secchi depth measurements of around 7% caused by deviations among individuals in their perception of contrast. Interestingly, there was no evidence of difference in skill between scientists and school children.

The classroom experiment demonstrates how the mini-Secchi disk may be used as an educational tool for teaching the concepts of aquatic optics, how light is attenuated with increasing depth and how different individuals perceive contrast. Other classroom-based experiments could easily be developed using the mini-Secchi disk. For instance, using different bins with different coloured dyes could help ascertain differences among individuals in their perception of colour, qualify deviations among individuals in FU colour measurements, and how the 7% difference in the Secchi depth recorded here from deviations among individuals might vary with water colour [59]. In contrast to teaching directly from textbooks, or through mathematical formulae, these types of active, hands-on teaching activities can provide deeper and more profound understanding of concepts [60]. Once more, the mini-Secchi disks used in such experiments could also be designed and manufactured in the classroom, using equipment available in most design and technology school departments (e.g., 3D printer, band saw and metal lathe). In fact, a batch of mini-Secchi disks were produced in the classroom at Chatham and Clarendon Grammar School as part of the REVIVAL project (see Figure 3 and https://pml.ac.uk/Research/Projects/REVIVAL). Actively designing, manufacturing and then using a scientific instrument in a planned hypothesis-driven experiment would be a powerful means of teaching the scientific method, and give students a grounding in the broad spectrum of skills in demand in aquatic optics.

#### 3.1.4. Temperature Measurements from the Mini-Secchi Disk

Comparison between the housed iButton and the NIST-traceable temperature probe (Figure 10b) on AMT28 indicate a median difference of 0.283 °C, within the manufacturers stated accuracy of the iButton (± 0.5 °C). This difference was relatively systematic over the temperature range tested (Figure 10b) and was therefore subtracted from all iButton data. Figure 10c indicates the corrected iButton data track tightly the continuous water temperature measurements from the underway CTD system on AMT28. Histogram of water temperature differences between the underway CTD and iButton (Figure 10d) show differences ≤0.05 °C, with little difference between predawn stations (median and mean absolute differences ∼0.03 °C) and noon stations (median absolute difference 0.03 °C and mean absolute difference of 0.06 °C). Histograms of water temperature differences between the profiler CTD and iButton (Figure 10e) also show median absolute differences of 0.05 °C (0.04 °C predawn, 0.08 °C noon), but higher mean differences (0.17 °C), owing to two outliers (not shown on the histogram at 1.19 and 1.72 °C), one occurring during a noon station the other during a predawn station).

The tests on AMT28 indicate it is possible to retrieve water temperature measurements with an accuracy of 0.05 °C using a calibrated iButton attached to the mini-Secchi disk. Without calibration, and according to manufacturers, this version of the iButton (the DS1922L Thermochron D/logger) has an accuracy of ±0.5 °C. For some coastal applications this is deemed reasonable, for other applications better accuracy is required. For example, space agencies target accuracy requirements for state-of-the-art satellite thermal sensors in coastal waters are <0.5 °C [62]. We envisage the accuracy of these miniature loggers will improve in the future. Meanwhile, for potential citizen science projects, the scientists could work with the citizens to do the calibrations or devise a simple method the citizens can use to do the calibrations themselves.

Water temperature is considered by the Global Climate Observing System as an Essential Climate Variable [63,64]. It controls aspects of the physical, biological and chemical state of aquatic water bodies [65] and has been linked to shifts in species and reconfigurations of aquatic communities [66,67,68,69]. Monitoring temperatures in lake and nearshore waters using the mini-Secchi disk could be useful for addressing a wide range of scientific questions, for instance on coupling between biological and physical processes [70,71], and for applications such as satellite validation [48,72] and environmental monitoring [73]. As environmental sensors become smaller, less expensive and more accurate, other important variables such as conductivity, pH and fluorescence could be measured by similar means using a mini-Secchi disk.

### 3.2. Future Directions for the Mini-Secchi Disk

Although it has already proven useful for various applications (Figure 7, Figure 8, Figure 9 and Figure 10), future improvements could be made to the mini-Secchi device. In its current form, approximately 7 to 8 m of tape can fit into the casing of the mini-Secchi device. Lottig et al. [24] found the average Secchi depth of 3251 lakes to be 2.4 m (median 2.1 m) with the vast majority of data <5 m (see Figure 3 of [24]). However, for some oligotrophic lakes, the Secchi depth can be as deep as 16 m [24]. Secchi depths can also regularly exceed 7 m in some nearshore regions, for example, around coral reefs [74].

Increasing the length of the tape could be achieved through a number of means, each with its caveats. The device could be made a little larger (including the disk), such that more tape could be fitted into the casing. Different sized devices could be developed for different conditions. The disadvantage of increasing the size is that the device may become less convenient to use. The tape could be made thinner, to enable more tape to fit into the casing. The disadvantage would be that the tape could lose some of its properties (e.g., resistance to wear and stretching), or become more expensive. One possible avenue to explore is to integrate mechanics within the device to log distance. One could then use a wire rather than a tape which would allow an increase in length (much smaller surface area) and also reduce drag on the tape when operating at higher current speeds. 3D printing mechanics within the casing is possible, but challenging. It would require precision printing and would likely reduce the durability of the device (more components vulnerable to failure). As 3D printers improve, further options to address this challenge may appear.

Although it is easy to use and provides a useful index of water colour, the vinyl-laminated FU colour scale sticker is not a perfect method for measuring water colour and future work should look to compare it with a standard FU scale. Variations in colour and durability (exposure to sunlight and water) of the sticker will depend on the quality of the vinyl printer used and its laminated finish. We recommend printing a number of these stickers and replacing them when sufficient wear dictates. The FU colour scale sticker is based simply on an RGB colour scale developed by Wernand et al. [19] for mapping the FU colour index using satellite data, and was designed to represents the FU colours “as close as possible” [19]. Improvements to this colour scale could be made in the future. In particular, the colour of water recorded by looking at a Secchi disk at half the Secchi depth can differ from that looking directly into the water (i.e., not using a disk) [75]. The colour scale could be improved to characterise this better, perhaps by taking photographs of the disk at half the Secchi depth, or through improved numerical radiative-transfer simulations that characterise better the effect of the background white disk on water colour. The mini-Secchi device could also benefit from the development of other do-it-yourself tools for measuring water colour based on the FU scale [76].

Though outside the scope of this work, in which the focus was on the production and demonstration of applications of the mini-Secchi device, data logging is a critical part of the measurement process. In addition to logging the Secchi depth and FU colour number, the location and the time are essential pieces of information. Other environmental data, for instance, sky conditions and wind speed, and are also useful. The use of mobile phone apps and online data entry forms have proven to aid in these types of data logging processes. Mobile apps for logging Secchi depth and FU colour number have been developed [25,28] as well as apps that glean information on water colour from mobile phone cameras [77,78,79]. These apps also make use of existing technology on the phone, such as GPS, and can instantaneously upload data onto servers for safe archiving. It is envisaged these types of mobile applications would be useful for logging mini-Secchi data.

Small, low cost electronics and embedded computing (e.g., https://www.arduino.cc/ and https://www.raspberrypi.org/) are opening new doors in environmental monitoring [80,81,82]. Low cost electronic tools are now being developed for citizen science projects with integrated sensors (e.g., GPS and environmental sensors) and bluetooth (or mobile upload) capabilities that minimise the effort required by the citizen to take the measurement, process and transfer the data (e.g., [49]). Future versions of the mini-Secchi device could benefit from these advances.

Future work should aim to characterise uncertainty in the measurement collected using the mini-Secchi disk. This is especially important for applications like satellite validation. This requires quantifying the measurement equation, deriving uncertainties in all components of the measurement equation, then propagating these uncertainties through the measurement equation, accounting for any correlations [83]. With regard to the mini-Secchi disk, some components influencing the measurement equation include: The reflectivity of the disk, the state of the water surface, the deployment platform (e.g., fixed or moving boat), the current speed (influencing the verticality of deployment), and the individuals perception of colour and contrast. Future work will need to focus on quantifying the uncertainties in all components for an appropriate characterisation of measurement uncertainty.

## 4. Summary

We present a simple hand-held, pocket-sized device, designed to measure water clarity (Secchi depth) and colour (FU scale) in lakes, estuaries and nearshore regions. This standardised device is constructed with marine resistant materials and manufactured using a 3D printer and some other basic workshop tools. It is light in weight, easy to use, and inexpensive to manufacture. We explain how the device was developed and provide detailed instructions on how to build it. These instructions are supported by electronic files. We illustrate how to maintain and use the device, and show how other sensors can be attached for measuring variables such as water temperature. A series of applications is presented, showing how the device can be used for measuring spatial variations in lake colour and clarity, how the size of the device makes it accessible to a range of water users previously not capable of measuring water colour and clarity, how it can be used to teach the concepts of aquatic optics, and how accurate temperature data can be collected by attaching a cheap, well-calibrated, temperature sensor to the weight of the device. We envisage future iterations of the device will benefit from improvements in 3D printing, improvements in the water colour scale used, mobile applications and small, low cost electronics.

## Figures and Tables

**Figure 1 sensors-19-00936-f001:**
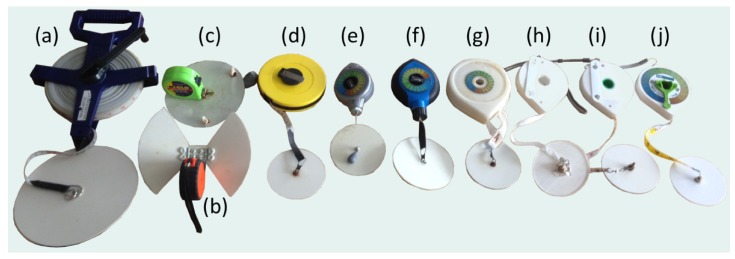
Mini-Secchi disk prototypes. (**a**) Traditional Secchi disk for use in lakes and nearshore waters (20 cm sized Secchi disk); (**b**) and (**c**) prototypes based on the use of retractable tape measures and foldable Secchi disks; (**d**–**f**) prototypes based on attaching a 10 cm disk to a series of small manually-operated tape measures and chalk lines; (**g**–**i**) initial 3D printed prototypes with 10 cm Secchi disks; and (**j**) the current 3D printed version of the mini-Secchi disk.

**Figure 2 sensors-19-00936-f002:**
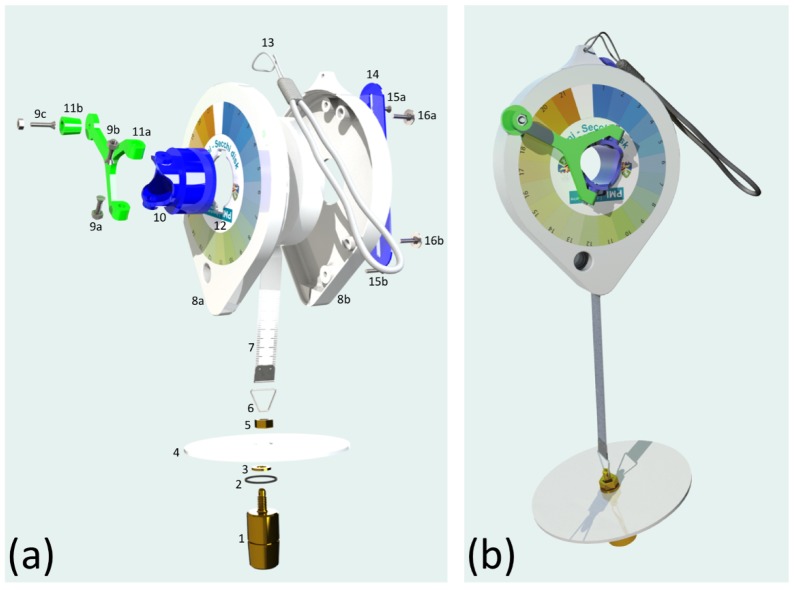
An exploded drawing (**a**) and rendering (**b**) of the mini-Secchi disk. Numbers refer to components (see Table 1 for description) and are listed as follows: 1. Weight; 2. Weight O-ring; 3. Weight washer; 4. Secchi disk; 5. Weight bolt nut; 6. Weight attachment circlip; 7. Tape measure; 8a,b. Mini-Secchi casing; 9a,b,c. Fixings; 10. Bobbin; 11a,b. Handle; 12. Colour scale; 13. Lanyard; 14. Finger strap; 15a,b. and 16a,b. Fixings.

**Figure 3 sensors-19-00936-f003:**
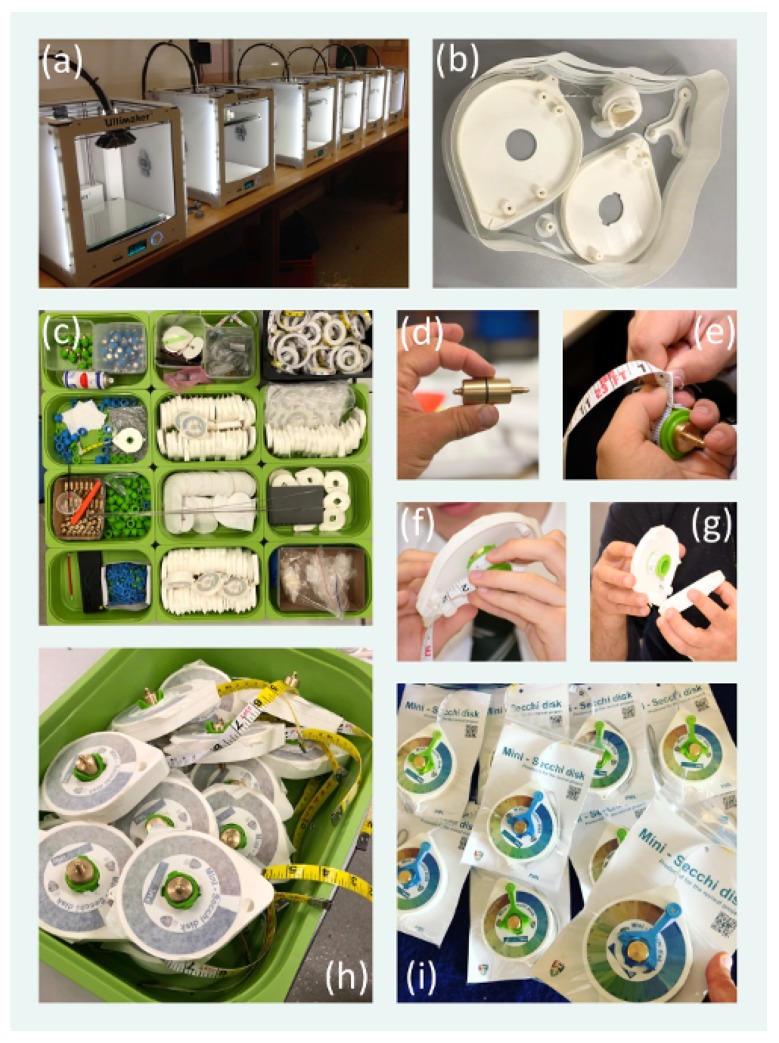
Photos of keys steps in the manufacturing process. (**a**) Ultimaker 2 printers used to 3D print components of the mini-Secchi disk. (**b**) An example 3D print of the casing, bobbin and handle of the mini-Secchi disk. (**c**) Assembly of the components of the mini-Secchi disk after 3D printing the parts and printing the vinyl colour scale sticker, manufacturing the weight and the Secchi disk, and procuring the measuring tape. (**d**–**g**) Constructing the mini-Secchi disk. Having manufactured the weight (**d**), using either a metal lathe or a CNC machine, and bobbin (glued and finished), the weight is fitted into the bobbin and the measuring tape is stitched onto the bobbin (**e**). The bobbin is fitted into the casing (**f**) and measuring tape enclosed between the two sides of the casing (**g**) before being locked in place with the finger strap using the fittings. (**h**) The colour scale sticker is added before the handle of the device is fitted, lanyard added, and disk and weight attached to the measuring tape. (**i**) Examples of mini-Secchi disks produced for the REVIVAL project (https://pml.ac.uk/Research/Projects/REVIVAL).

**Figure 4 sensors-19-00936-f004:**
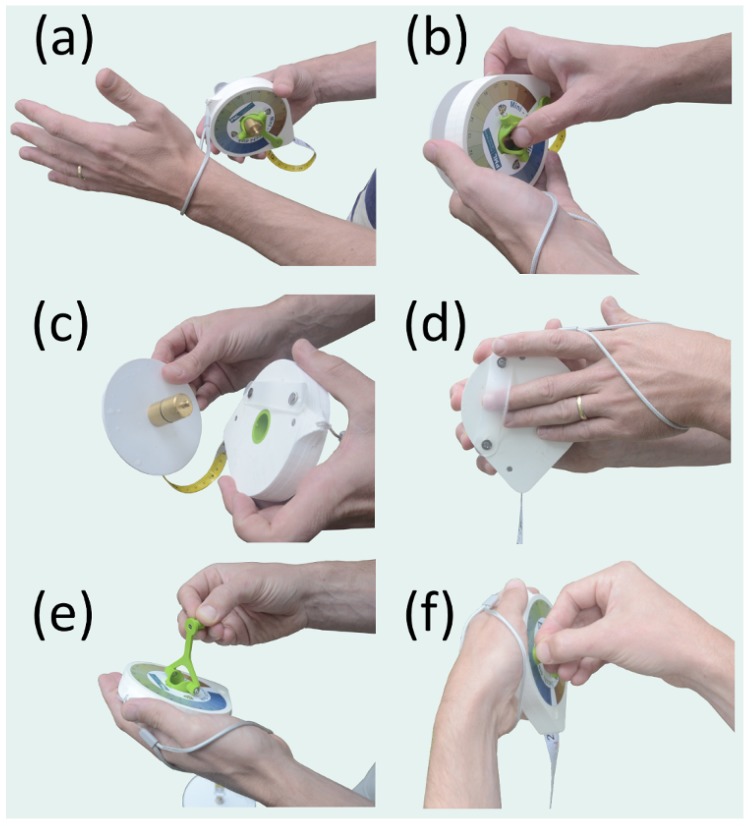
Operating the mini-Secchi disk. (**a**) Lanyard is slipped over the wrist and (**b**) weight pushed out from the bobbin. (**c**) Disk detached and (**d**) holding hand fingers slipped under the finger strap. (**e**) Handle is detached from the casing and flipped 180 degrees. (**f**) Handle is used to wind the tape measure in and out of the main body of the mini-Secchi disk.

**Figure 5 sensors-19-00936-f005:**
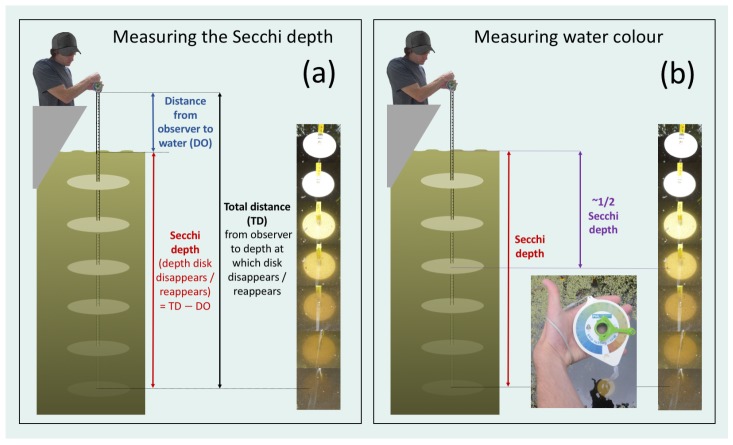
Measuring Secchi depth and water colour with the mini-Secchi disk. (**a**) Measuring the Secchi depth. (**b**) Measuring water colour at half the Secchi depth.

**Figure 6 sensors-19-00936-f006:**
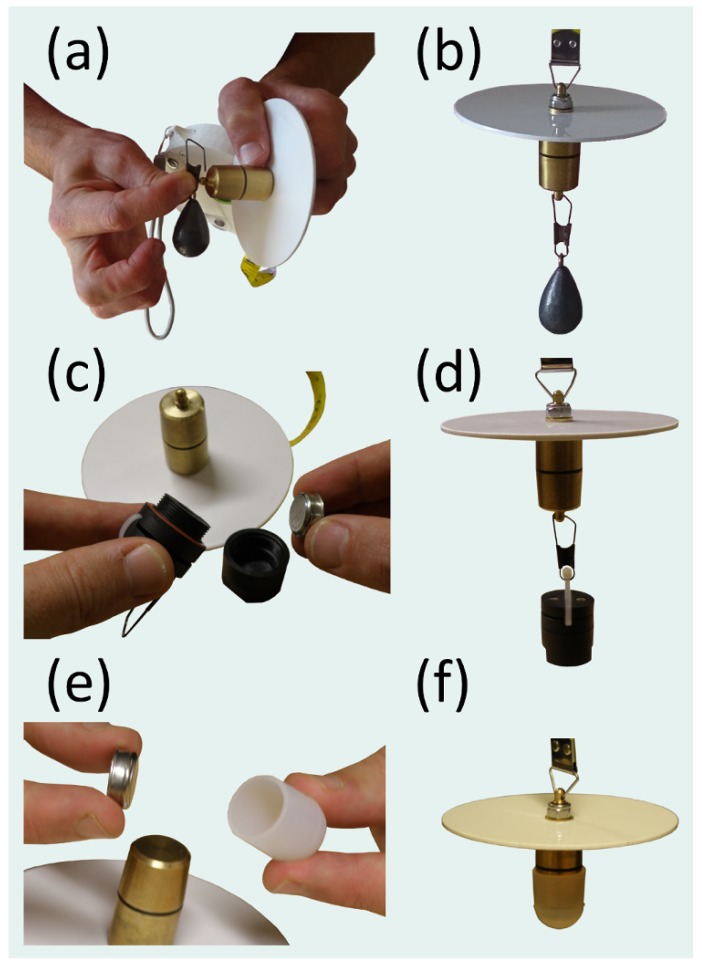
Additional weight and sensors can be added to the mini-Secchi disk. (**a**–**b**) Shows how additional weight (a standard fishing weight) can be attached to one of the weight designs of the mini-Secchi disk for operation in currents, or in cases where the platform may be moving (e.g., from a boat). (**c**–**d**) An iButton housed in a Thermochron waterproof capsule (DS9107) is attached to one of the weight designs of the mini-Secchi disk for measuring water temperature. (**e**–**f**) An iButton housed in a Thermochron sinking silicon capsule (Th-Silenc) is attached to the weight of the mini-Secchi disk for measuring water temperature.

**Figure 7 sensors-19-00936-f007:**
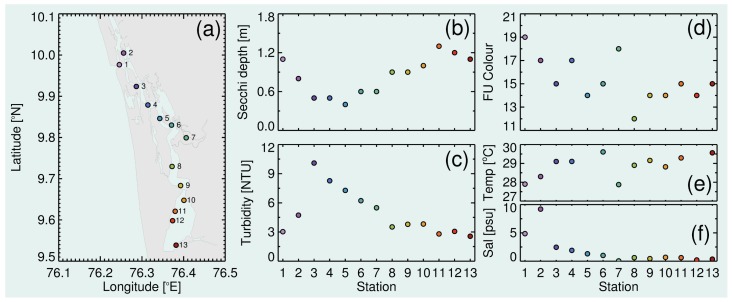
Deployment of the mini-Secchi disk in Vembanad lake on the 30th and 31st May 2018. (**a**) Stations sampled with the mini-Secchi disk and CTD rosette sampler equipped with a WET Labs ECO triplet. (**b**) Secchi depth readings using the mini-Secchi disk, (**c**) water turbidity at the surface (top 1 m), and (**d**) Forel Ule colour readings using the mini-Secchi disk. (**e**) Surface (top 1 m) temperature (Temp) and (**f**) salinity (Sal) for the 13 stations sampled.

**Figure 8 sensors-19-00936-f008:**
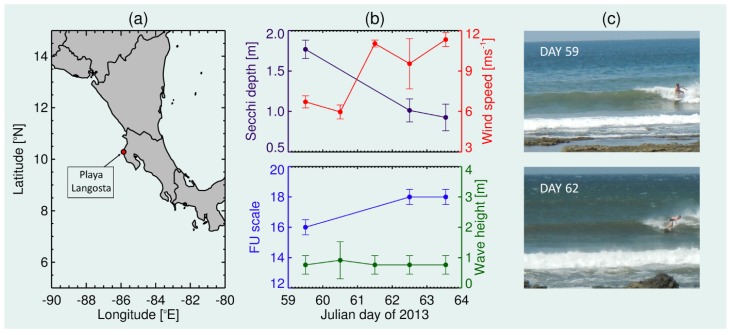
Deployment of the mini-Secchi disk by a surfer. (**a**) Location of Playa Langosta in Costa Rica. (**b**) Data collected using the mini-Secchi disk (Secchi-depth and FU colour scale) between 28th February and 4th March 2013 at Playa Langosta. Wind speed data from nearby weather station (Sede de Guanacaste, Liberia) and wave height data at Langosta (using wave model forced with observations) are also shown. Uncertainties (error bars) for the Secchi depth and wind speed are based on standard deviations of measurements during the surfing session (5–7 for the Secchi depth), or around local noon for wind speed for the days of no deployment. Uncertainties (error bars) in wave height represent the range in wave heights from the wave model data. Uncertainties (error bars) in FU colour scale were set to half an FU number based on reproducibility results in Wernand and van der Woerd [5]. (**c**) Photos taken of the surfer at Playa Langosta in Costa Rica on the 28th February (Day 59) and 3th March (Day 62).

**Figure 9 sensors-19-00936-f009:**
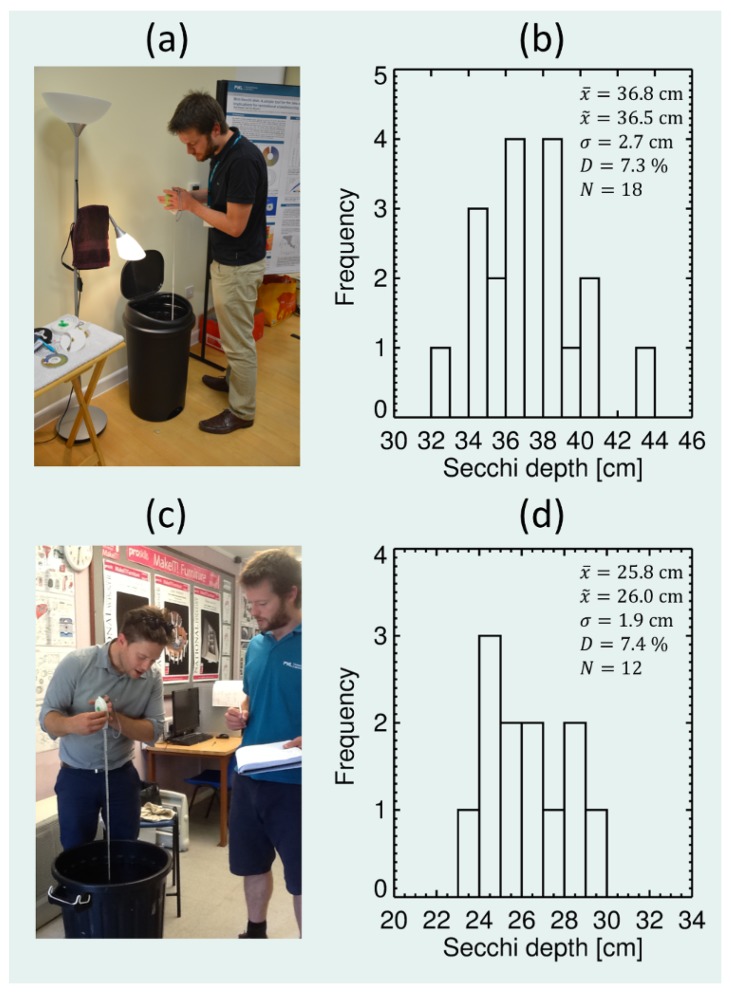
Quantifying uncertainty arising from different individuals reading the Secchi depth. The experimental set up consisted of a mini-Secchi disk and a bin filled with water and food colouring. (**a**) Experimental set-up at the Challenger Society and Remote Sensing and Photogrammetry Society Marine Optics Special Interest Group (MOSIG) on the 16th December 2013. (**b**) Histogram of Secchi depth readings taken at the MOSIG meeting. (**c**) Experimental set-up during a design and technology lesson at a secondary school at Chatham and Clarendon Grammar School on the 4th July 2014. (**d**) Histogram of Secchi depth readings during the design and technology lesson. $\bar{x}$ is the average Secchi depth, $\tilde{x}$ is the median Secchi depth, σ is the standard deviation of the Secchi depth readings, *D* is the percentage deviation (σ/$\bar{x}\times 100$) and *N* is the number of measurements.

**Figure 10 sensors-19-00936-f010:**
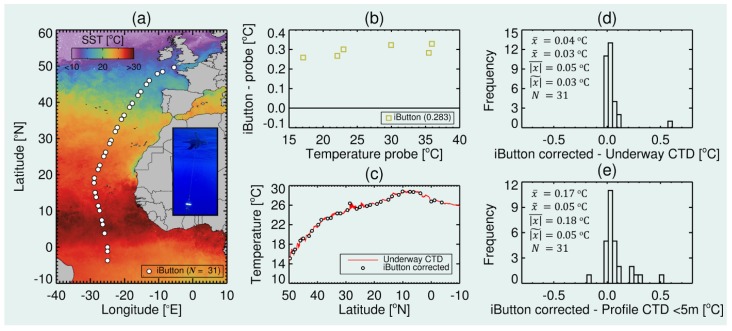
Evaluating the accuracy of water temperature measurements from an iButton attached to the mini-Secchi disk. (**a**) Deployments locations of the Small Oceanographic floating Device (SOD, see photo) with iButton and mini-Secchi disk on AMT28, overlain onto a Sea Surface Temperature (SST) monthly composite of October 2018 from the MODIS-Aqua satellite [61]. (**b**) Results from the comparison between the housed iButton and the NIST-traceable probe at different temperatures in a recirculating water bath. The value 0.283 represents median difference in temperature (°C). (**c**) Along-track SST from the underway CTD system with the iButton data (corrected following removal of systematic difference of 0.283 °C) overlain. (**d**) Histogram of the water temperature differences between the underway CTD and iButton on the mini-Secchi disk. (**e**) Histogram of the water temperature differences between the profiler CTD in the top 5 m (upcast) and the iButton on the mini-Secchi disk. Two outliers are not shown on the graph (1.19 and 1.72 °C). $\bar{x}$ is the average difference, $\tilde{x}$ is the median difference, $\bar{\left|x\right|}$ is the average absolute difference, $\tilde{\left|x\right|}$ is the median absolute difference and *N* is the number of samples.

**Table 1 sensors-19-00936-t001:** Components of the mini-Secchi disk.

Name	Figure 2a Index	Description	Materials	Manufacture or Purchase Method	Dimensions
Weight	1	To ensure the Secchi disk sinks vertically.	We tested various materials and eventually chose marine grade brass as it is very dense and to minimise corrosion when operated in the marine environment. The weight is 100 g. Two designs are provided. One includes an attachment to allow extra weight to be added, when operated in currents, and for attaching additional sensors, for example, a temperature sensor (see Figure 6).	For single production the weight can be produced (home-made) using a metal lathe (see dimensions of weight in Appendix A). For batch production, the weight can be computer numerical control (CNC) machined by any local CNC company from the STL file (see Appendix A; we have used G&J Services Ltd in Kent in the past (http://www.gandjcnc.co.uk) for some of our batch runs). It is also possible to 3D print the weight using a suitable metal filament and appropriate 3D printer. If you try this you want to achieve a weight of 100 g or heavier (we would be interested to know how you get on).	Technical drawing and STL file for both designs are provided in Appendix A.
Weight O-ring	2	To ensure weight locks into bobbin.	Rubber.	Purchasable from gasket and bearing supplier.	Provided in Appendix A.
Weight washer	3	Separates disk from weight.	Brass or stainless steel.	Purchasable from a hardware store.	Standard M6 washer (12 mm outside diameter, 6.4 mm centre hole diameter and thickness 1.6 mm).
Secchi Disk	4	White Secchi disk.	Polypropylene. Very high resistance to UV. An oily plastic with good elastic properties. Resistant to becoming brittle and capable of retaining its colour over time.	Polypropylene sheet purchasable from hardware store or online plastic distributor. Can be manufactured at home or in a workshop using a compass to mark outer circle at 100 mm diameter, then cut using a bandsaw or by hand using coping saw. 6 mm hole drilled in the center. Can also be produced in batch using a laser cutter.	100 mm diameter, 2 mm thick and 6 mm central hole.
Weight bolt nut	5	To secure weight to Secchi disk.	M6 brass or stainless steel nut with ’nyloc’ or nylon locking insert incorporated.	Purchasable from hardware store.	Standard M6 ‘nyloc’ nut.
Weight attachment circlip	6	Joins tape to the weight and Secchi disk. This must be bent to allow the start of the tape measure to be at the correct distance from the disk so the measurement is accurate (tape starts at ∼20 mm from disk).	Stainless steel.	0.7 mm stainless steel rod cut to 30 mm with pliers, bent using a simple jig set into a 10 mm by 10 mm by 10 mm equilateral triangle and threaded onto the weight and stainless steel fastening on the end of the tape. We bought this from an online hardware store. Alternatively, you can buy a triangular stainless 0.7 mm corner clip from a hardware store and adjust it with long nose pliers and pincers to the correct length. Ensure circlip is strong enough to carry a sufficient load (in strong currents extra weight is required to sink disk vertically and there can be drag on the disk).	10 mm by 10 mm by 10 mm equilateral triangle that is 0.7 mm thick.
Tape measure	7	Tape used to measure the Secchi depth (printed in imperial and metric units). Stitched to bobbin at one end and attached to weight at other end by stainless steel fastening.	Tape made from fibre-glass reinforced polypropylene. Fibre-glass tape makes it resistant to stretching and polypropylene material makes it resistance to wear in aquatic environments. Nylon thread and needle required for stitching tape to bobbin. Stainless steel fastening.	Tape purchasable from on-line provider or can be extracted from a standard fibre-glass tape measure purchased from hardware store or online (we have used the 10 m senator fibreglass tape measure from Cromwell tools (http://www.cromwell.co.uk) in the past). Stainless steel sheet (for fastening) purchased from hardware store and riveted onto tape at a distance of 20 mm into the tape using a rivet tool (20 mm distance allowing for circlip, weight bolt and weight nut).	13 mm wide tape. Length of tape is dependent on size of casing and thickness of tape. The typical length used is 7 to 8 m. Stainless steel sheet 30 mm length, 13 mm width, and 0.2 mm thick. Two stainless steel 3.2 mm rivets.
Mini-Secchi casing	8a,b	Structural casing of the mini-Secchi disk. Framework encasing the measuring tape and bobbin.	The casing is made from polylactic acid. This is a biodegradable thermoplastic devised from renewable resources or natural starch. Modern polylactic acid plastic has good structural properties and when deposed can be composted.	3D printed using the Ultimaker 2 and 2+. Any 3D printer would work for this manufacturing process as the files are available in STL format. We used the Ultimaker as it is a robust, low-cost printer, and the software is open source and managed well by the company. We have clocked up almost four years of solid run time and these printers are still operating like new. Ultimaker Cura software allows for the nesting of components to make the printing process as efficient as possible. Once printed all parts require a post-print clean up. Depending on the quality of the print you have achieved will determine the clean up required. For FDM printers, a set of modelling making chisels are useful for the task.	STL file provided in Appendix A.
Fixings 1	9a,b,c	Bolts and nuts to fix handle to bobbin.	Stainless steel.	Purchasable from hardware shop.	M2.5 nuts and bolts. Bolts are 10 mm long and have a countersunk flat screw head.
Bobbin	10	Cylinder holding the tape and for storing the weight. Rotates to wind tape in and out of casing.	Polylactic acid biodegradable thermoplastic (see mini-Secchi casing for details).	3D printed using the Ultimaker 2 and 2+ (see mini-Secchi casing for details). Two parts printed separately. In order to have a very accurate fit for the brass weight we printed the bobbin with a high-quality fill and finish. The two parts were glued together with an industrial super glue using a 19 mm bar inserted into the hole in the middle of each component to insure parts were accurately aligned. 20 mm reamer bit used to finish inside hole for accurate fit to brass weight (either by mounting reamer bit in the jaws of a metal work lathe and sliding the bobbin over from end to end, or reaming by hand using a decent pair of gloves).	STL files provided in Appendix A.
Handle	11a,b	Handle used to wind tape in and out of casing. Two separate components to allow for efficient rotation. Fixed together with a stainless steel nut and bolt (see Fixings 1).	Polylactic acid biodegradable thermoplastic (see mini-Secchi casing for details)	3D printed using the Ultimaker 2 and 2+ (see mini-Secchi casing for details).	STL files provided in Appendix A.
Colour scale	12	Vinyl-laminated Forel Ule colour scale sticker. Red Green Blue (RGB) colours for each Forel Ule colour were taken from Wernand et al. [19], see their Table 5.	Synthetic textile fibre (long-chain polymer) consisting of vinyl alcohol units.	Printed using a vinyl printer.	Example vinyl print file provided in Appendix A.
Lanyard	13	Standard camera wrist lanyard for carrying the device and for preventing the device from dropping when in use.	Nylon.	Purchasable from a camera shop or online.	180 mm length.
Finger strap	14	Used to hold the device safely when in operation (by slipping one or two fingers under the strap).	0.5 mm polypropylene. Very high resistance to UV. An oily plastic with good elastic properties. High resistance to becoming brittle and capable of retaining its colour over time.	Technical drawing can be overlain onto polypropylene sheet as a template (stuck down using masking tape or spray mount) and cut out with scissors or a craft knife. Can also be laser cut from the template.	Technical drawing provided in Appendix A.
Fixings 2	15a,b	Bolts to fix casing together. These screw directly into the 3D printed casing.	Stainless steel.	Purchasable from hardware shop.	M2.5 nuts and bolts. Bolts are 10 mm long and have a countersunk flat screw head.
Fixings 3	16a,b	Bolts, washers and nuts to fix finger strap to casing. Nuts fit into body of the chassis.	Stainless steel.	Purchasable from hardware shop.	M2.5 nuts, washers and bolts. Bolts are 10 mm long and have a countersunk flat screw head.

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
