# Peer review of "A Printable Device for Measuring Clarity and Colour in Lake and Nearshore Waters"

_sensors, 2019, doi:10.3390/s19040936_

Round 1

Reviewer 1 Report

The manuscript describes the development, possible uses, and future improvements of a 3D-printed device consisting of a mini Secchi disk that integrates an FU scale. The description of the device is very accurate, and its applications are well documented.
Considering the number of applications highlighted by the authors, the advantages introduced by the specific design of the device, and availability of all the necessary 3D models, I see this work as a relevant addition to the literature concerning citizen science applied to ocean colour, and I recommend its publication.

I have only a little concern which I would like the authors to comment on.

The benefit induced by citizen science in terms of increasing people’s awareness and improving in-situ data coverage is evident, anyway the measurement uncertainty assessment (and thus the consequent quality control) is still an issue.
In this sense, the mini Secchi is an extraordinary educational tool as well as a valid instrument to perform a general assessment of water clarity and colour, especially for the estimation of temporal and spatial trends through relative measurements. It is also a good hosting platform for other sensors, as outlined in section 3.1.4. At the same time, in order to be used for satellite validation, a comprehensive evaluation of the measurement uncertainty is still missing. This should be stated in the manuscript. Additionally, to avoid misleading interpretations, I would also suggest replacing the term “uncertainty” in section 3.1.3 with “percent deviation”.  
Alternatively, I strongly encourage the authors to include a section about measurement uncertainty. This could benefit from the results of previous studies applied to standard Secchi disks and include the percent deviation presented in section 3.1.3, together with other measurements in a real environment.

Author Response

Please find our response in the attached PDF file

Reviewer 2 Report

I have read the manuscript with great interest. The enthusiasm of the authors in developing and testing the device is clearly visible in all pages. Even measurements during a honeymoon of one of the authors were performed. Also, like we know from other articles from this first author, the storyline, text, figures and references are all of high quality. No major mistakes are made and I only have a few minor remarks.

However, last days I wondered if it is innovative enough to be published. There are hundreds of articles on Secchi disk readings and some more on the Forel-Ule colour index. Also, I can imagine that the whole process of 3D printing is developing so quickly, that soon this method will be taken for granted.

On the website of Sensors it is stressed that: ‘Manuscripts regarding research proposals and research ideas are particularly welcome’. I am convinced that that this manuscript fits this description. It has the very strong point that after each demonstration a detailed explanation is given how this will help the education, coastal monitoring and environmental awareness of large groups of citizens. It is the combination of a new “sensor”, 3D printing and outreach to new communities that make it worth to publish.

Finally, a word on testing and development in class. In the Citclops project it became clear that as soon as electronics or complicated assembly procedures were required (KdUINO, SmartFluo) even high schools in Spain and Germany could not cope: it took too much time for teachers to understand and guide. So, these instruments were developed and tested by very specific students (first class technical universities) in small numbers.

Minor comments:

L12 “operation”

L40 “ US, based”

Fig. 2. Mention the diameter of the casing.

Table 1 now covers 3 pages and goes into great detail. Maybe better to move it to the supplementary material.

L186. Give brand of nephelometer to make sure it is a reliable validation measurements.

L214. ‘dissolvable food colouring’. Please make a proper reference and indicate the colour. As the authors are well aware, the work by ZP Lee proofs that the colour makes a difference.

Fig. 7. Personally, I find the blue backdrop of these figures not OK. Please remove.

Fig. 8. Please provide the number of independent observations (N=??). Also, these observations were made more than 5 years ago and probably not with the min-disk as presented here. Please explain.

L351, 352. No all citizens have access to the AMT28 facilities. That you have nicely calibrated the bias in your iButton is not the issue. The problem is if all iButtons have a bias?  Is that bias the same? How can citizens check that. We are talking about minute differences that might not be problematic for coastal and lake measurements.

L 381-394 The use of a printed colour scale is not validated in this manuscript. I admit that this would a major exercise that hopefully might take place in the future. On its own it already good that citizen scientist are aware of colour and take the trouble to approximate this colour by a rough print that can be easily compared to the colour above the Secchi Disk.

Author Response

Please find our response to comments in the attached PDF file.

Reviewer 3 Report

This manuscript details the development and use of a mini secchi disk designed to be easily used by citizen scientists. This is a rather novel and well-implemented design. The authors have tested the secchi disk in various locations and it was found to be easy to use. Also addressed are some of the possible limitations and improvement. I find this well-written paper with no major edits required. The authors have provided all the necessary information for people to make this device for themselves.

Author Response

Please find our response in the attached PDF file.
